# MEMORY-BASED GRAPH NETWORKS

**Amir Hosein Khasahmadi**[1,2] [*], **Kaveh Hassani**[3], **Parsa Moradi**[4], **Leo Lee**[1,2], **Quaid Morris**[1,2]

[1]University of Toronto, Toronto, Canada          [2]Vector Institute, Toronto, Canada
[3]Autodesk AI Lab, Toronto, Canada          [4]Sharif University of Technology, Tehran, Iran
{amirhosein,ljlee}@psi.toronto.edu          kaveh.hassani@autodesk.com
parsa.mordadi@ee.sharif.edu          quaid.morris@utoronto.ca

## ABSTRACT

Graph neural networks (GNNs) are a class of deep models that operate on data with arbitrary topology represented as graphs. We introduce an efficient memory layer for GNNs that can jointly learn node representations and coarsen the graph. We also introduce two new networks based on this layer: memory-based GNN (MemGNN) and graph memory network (GMN) that can learn hierarchical graph representations. The experimental results show that the proposed models achieve state-of-the-art results in eight out of nine graph classification and regression benchmarks. We also show that the learned representations could correspond to chemical features in the molecule data. Code and reference implementations are released at: https://github.com/amirkhas/GraphMemoryNet

## 1 INTRODUCTION

Graph neural networks (GNNs) (Wu et al., 2019; Zhou et al., 2018; Zhang et al., 2018) are a class of deep models that operate on data with arbitrary topology represented as graphs such as social networks (Kipf & Welling, 2016), knowledge graphs (Vivona & Hassani, 2019), molecules (Duvenaud et al., 2015), point clouds (Hassani & Haley, 2019), and robots (Wang et al., 2019). Unlike regular-structured inputs such as grids (e.g., images and volumetric data) and sequences (e.g., speech and text), GNN inputs are permutation-invariant variable-size graphs consisting of nodes and their interactions. GNNs such as gated GNN (GGNN) (Li et al., 2015), message passing neural network (MPNN) (Gilmer et al., 2017), graph convolutional network (GCN) (Kipf & Welling, 2016), and graph attention network (GAT) (Velikovi et al., 2018) learn node representations through an iterative process of transferring, transforming, and aggregating the node representations from topological neighbors. Each iteration expands the receptive field by one hop and after $k$ iterations the nodes within $k$ hops influence the node representations of one another. GNNs are shown to learn better representations compared to random walks (Grover & Leskovec, 2016; Perozzi et al., 2014), matrix factorization (Belkin & Niyogi, 2002; Ou et al., 2016), kernel methods (Shervashidze et al., 2011; Kriege et al., 2016), and probabilistic graphical models (Dai et al., 2016).

These models, however, cannot learn hierarchical representations as they do not exploit the compositional nature of graphs. Recent work such as differentiable pooling (DiffPool) (Ying et al., 2018), TopKPool (Gao & Ji, 2019), and self-attention graph pooling (SAGPool) (Lee et al., 2019) introduce parametric graph pooling layers that allow GNNs to learn hierarchical graph representations by stacking interleaved layers of GNN and pooling layers. These layers cluster nodes in the latent space. The clusters may correspond to communities in a social network or potent functional groups within a chemical dataset. Nevertheless, these models are not efficient as they require an iterative process of message passing after each pooling layer.

In this paper, we introduce a memory layer for joint graph representation learning and graph coarsening that consists of a multi-head array of memory keys and a convolution operator to aggregate the soft cluster assignments from different heads. The queries to a memory layer are node representations from the previous layer and the outputs are the node representations of the coarsened graph. The memory layer does not explicitly require connectivity information and unlike GNNs

---

[*]Work done during internship at Autodesk Toronto AI Lab.

relies on the global information rather than local topology. Hence, it does not struggle with over-smoothing problem (Xu et al., 2018; Li et al., 2018). These properties make memory layers more efficient and improves their performance. We also introduce two networks based on the proposed layer: memory-based GNN (MemGNN) and graph memory network (GMN). MemGNN consists of a GNN that learns the initial node representations, and a stack of memory layers that learn hierarchical representations up to the global graph representation. GMN, on the other hand, learns the hierarchical representations purely based on memory layers and hence does not require message passing.

## 2 RELATED WORK

**Memory augmented neural networks (MANNs)** utilize external memory with differentiable read-write operators allowing them to explicitly access the past experiences and are shown to enhance reinforcement learning (Pritzel et al., 2017), meta learning (Santoro et al., 2016), few-shot learning (Vinyals et al., 2016), and multi-hop reasoning (Weston et al., 2015). Unlike RNNs, in which the memory is represented within their hidden states, the decoupled memory in MANNs allows them to store and retrieve longer term memories with less parameters. The memory can be implemented as: *key-value memory* such as neural episodic control (Pritzel et al., 2017) and product-key memory layers (Lample et al., 2019), or *array-structured memory* such as neural Turing machine (NTM) (Graves et al., 2014), prototypical networks (Snell et al., 2017), memory networks (Weston et al., 2015), and sparse access memory (SAM) (Rae et al., 2016). Our memory layer consists of a multi-head array of memory keys.

**Graph neural networks (GNNs)** mostly use message passing to learn node representations over graphs. GraphSAGE (Hamilton et al., 2017) learns representations by sampling and aggregating neighbor nodes whereas GAT (Velikovi et al., 2018) uses attention mechanism to aggregate representations from all neighbors. GCN extend the convolution operator to arbitrary topology. Spectral GCNs (Bruna et al., 2014; Defferrard et al., 2016; Kipf & Welling, 2016) use spectral filters over graph Laplacian to define the convolution in the Fourier domain. These models are less efficient compared to spatial GCNs (Schlichtkrull et al., 2018; Ma et al., 2019) which directly define the convolution on graph patches centered on nodes. Our memory layer uses a feed-forward network to learn the node representations.

**Graph pooling** can be defined in global or hierarchical manners. In former, node representations are aggregated into a graph representation by a readout layer implemented using *arithmetic operators* such as summation or averaging (Hamilton et al., 2017; Kipf & Welling, 2016) or *set neural networks* such as Set2Set (Vinyals et al., 2015) and SortPool (Morris et al., 2019). In latter, graphs are coarsened in each layer to capture the hierarchical structure. Efficient non-parametric methods such as clique pooling (Luzhnica et al., 2019), kNN pooling (Wang et al., 2018), and Graclus (Dhillon et al., 2007) rely on topological information but are outperformed by parametric models such as edge contraction pooling (Diehl, 2019).

DiffPool (Ying et al., 2018) trains two parallel GNNs to compute node representations and cluster assignments using a multi-term loss including classification, link prediction, and entropy losses, whereas Mincut pool (Bianchi et al., 2019) trains a sequence of a GNN and an MLP using classification loss and the minimum cut objective. TopKPool (Cangea et al., 2018; Gao & Ji, 2019) computes node scores by learning projection vectors and dropping all the nodes except the top scoring nodes. SAGPool (Lee et al., 2019) extends the TopKPool by using graph convolutions to take neighbor node features into account. We use a clustering-friendly distribution to compute the attention scores between nodes and clusters.

## 3 METHOD

### 3.1 MEMORY LAYER

We define a memory layer $\mathcal{M}^{(l)} : \mathbb{R}^{n_l \times d_l} \longmapsto \mathbb{R}^{n_{l+1} \times d_{l+1}}$ in layer $l$ as a parametric function that takes in $n_l$ query vectors of size $d_l$ and generates $n_{l+1}$ query vectors of size $d_{l+1}$ such that $n_{l+1} < n_l$. The input and output queries represent the node representations of the input graph and the coarsened graph, respectively. The memory layer learns to jointly coarsen the input nodes,

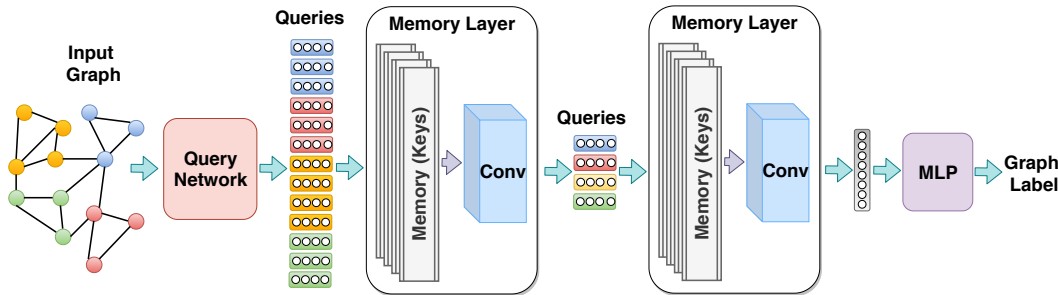

Figure 1: The proposed architecture for hierarchical graph representation learning using the proposed memory layer. The query network projects the initial node features into a latent query space and each memory layer jointly coarsens the input queries and transforms them into a new query space.

i.e., pooling, and transform their features, i.e., representation learning. As shown in Figure 1, a memory layer consists of arrays of memory keys, i.e., multi-head memory, and a convolutional layer. Assuming $|h|$ memory heads, a shared input query is compared against all the keys in each head resulting in $|h|$ attention matrices which are then aggregated into a single attention matrix using the convolution layer.

In a *content addressable memory* (Graves et al., 2014; Sukhbaatar et al., 2015; Weston et al., 2015), the task of attending to memory, i.e., addressing scheme, is formulated as computing the similarity between memory keys to a given query $q$. Specifically, the attention weight of key $k_j$ for query $q$ is defined as $w_j = softmax(d(q, k_j))$ where $d$ is a similarity measure, typically Euclidean distance or cosine similarity (Rae et al., 2016). The soft read operation on memory is defined as a weighted average over the memory keys: $r = \sum_j w_j k_j$.

In this work, we treat the input queries $\mathbf{Q}^{(l)} \in \mathbb{R}^{n_l \times d_l}$ as the node representations of an input graph and treat the keys $\mathbf{K}^{(l)} \in \mathbb{R}^{n_{l+1} \times d_l}$ as the cluster centroids of the queries. To satisfy this assumption, we impose a clustering-friendly distribution as the distance metric between keys and a query. Following (Xie et al., 2016; Maaten & Hinton, 2008), we use the Student's t-distribution as a kernel to measure the normalized similarity between query $q_i$ and key $k_j$ as follows:

$$C_{i,j} = \frac{\left(1 + ||q_i - k_j||^2/\tau\right)^{-\frac{\tau+1}{2}}}{\sum\limits_{j'} \left(1 + ||q_i - k_{j'}||^2/\tau\right)^{-\frac{\tau+1}{2}}} \tag{1}$$

where $C_{ij}$ is the normalized score between query $q_i$ and key $k_j$, i.e., probability of assigning node $i$ to cluster $j$ or attention score between query $q_i$ and memory key $k_j$, and $\tau$ is the degree of freedom of the Student's t-distribution, i.e., temperature. To increase the capacity, we model the memory keys as a multi-head array. Applying a shared input query against the memory keys produces a tensor of cluster assignments $\left[\mathbf{C}_0^{(l)} ... \mathbf{C}_{|h|}^{(l)}\right] \in \mathbb{R}^{|h| \times n_{l+1} \times n_l}$ where $|h|$ denotes the number of heads. To aggregate the heads into a single assignment matrix, we treat the heads and the assignments matrices as depth, height, and width channels in standard convolution analogy and apply a convolution operator over them. Because there is no spatial structure, we use $[1 \times 1]$ convolution to aggregate the information across heads and therefore the convolution behaves as a weighted pooling that reduces the heads into a single matrix. The aggregated assignment matrix is computed as follows:

$$\mathbf{C}^{(l)} = \text{softmax}\left(\Gamma_\phi\left(\mathop{\Big\|}_{k=0}^{|h|} \mathbf{C}_k^{(l)}\right)\right) \in \mathbb{R}^{n_l \times n_{l+1}} \tag{2}$$

where $\Gamma_\phi$ is a $[1 \times 1]$ convolutional operator parametrized by $\phi$, $||$ is the concatenation operator, and $\mathbf{C}^{(l)}$ is the aggregated soft assignment matrix.

A memory read generates a value matrix $\mathbf{V}^{(l)} \in \mathbb{R}^{n_{l+1} \times d_l}$ that represents the coarsened node representations in the same space as the input queries and is defined as the product of the soft assignment scores and the original queries as follows:

$$\mathbf{V}^{(l)} = \mathbf{C}^{(l)\top} \mathbf{Q}^{(l)} \in \mathbb{R}^{n_{l+1} \times d_l} \tag{3}$$

The value matrix is fed to a single-layer feed-forward neural network to project the coarsened embeddings from $\mathbb{R}^{n_{l+1} \times d_l}$ into $\mathbb{R}^{n_{l+1} \times d_{l+1}}$ as follows:

$$\mathbf{Q}^{(l+1)} = \sigma \left( \mathbf{V}^{(l)} \mathbf{W} \right) \in \mathbb{R}^{n_{l+1} \times d_{l+1}} \tag{4}$$

where $\mathbf{Q}^{(l+1)} \in \mathbb{R}^{n_{l+1} \times d_{l+1}}$ is the output queries, $\mathbf{W} \in \mathbb{R}^{d_l \times d_{l+1}}$ is the network parameters, and $\sigma$ is the non-linearity implemented using LeakyReLU.

Thanks to these parametrized transformations, a memory layer can jointly learn the node representations and coarsens the graph end-to-end. The computed queries $\mathbf{Q}^{(l+1)}$ are the input queries to the subsequent memory layer $\mathcal{M}^{(l+1)}$. For graph classification tasks, one can simply stack layers of memory up to the level where the input graph is coarsened into a single node representing the global graph representation and then feed it to a fully-connected layer to predict the graph class as follows:

$$\mathcal{Y} = \text{softmax} \left( \text{MLP} \left( \mathcal{M}^{(l)} \left( \mathcal{M}^{(l-1)} \left( ....\mathcal{M}^{(0)} \left( \mathbf{Q}_0 \right) \right) \right) \right) \right) \tag{5}$$

where $\mathbf{Q}_0 = f_q(g)$ is the initial query representation[1] generated by the query network $f_q$ over graph $g$. We introduce two architectures based on the memory layer: GMN and MemGNN. These two architectures are different in the way that the query network is implemented. More specifically, GMN uses a feed-forward network for initializing the query: $f_q(g) = \text{FFN}_\theta(g)$, whereas MemGNN implements the query network as a message passing GNN: $f_q(g) = \text{GNN}_\theta(g)$.

## 3.2 GMN ARCHITECTURE

A GMN is a stack of memory layers on top of a query network $f_q(g)$ that generates the initial query representations without any message passing. Similar to set neural networks (Vinyals et al., 2015) and transformers (Vaswani et al., 2017), nodes in GMN are treated as a permutation-invariant set of representations. The query network projects the initial node features into a latent space that represents the initial query space.

Assume a training set $\mathcal{D} = [g_1, g_2, ..., g_N]$ of $N$ graphs where each graph is represented as $g = (\mathbf{A}, \mathbf{X}, Y)$ and $\mathbf{A} \in \{0, 1\}^{n \times n}$ denotes the adjacency matrix, $\mathbf{X} \in \mathbb{R}^{n \times d_{in}}$ is the initial node features, and $Y \in \mathbb{R}^n$ is the graph label. Considering that GMN treats a graph as a set of order-invariant nodes and does not use message passing, and also considering that the memory layers do not rely on connectivity information, the topological information of each node should be somehow encoded into its initial representation. To define the topological embedding, we use an instantiation of general graph diffusion matrix $\mathbf{S} \in \mathbb{R}^{n \times n}$. More specifically, we use random walk with restart (RWR) (Pan et al., 2004) to compute the topological embeddings and then sort them row-wise to force the embedding to be order-invariant. For further details please see section A.4. Inspired by transformers (Vaswani et al., 2017), we then fuse the topological embeddings with the initial node features into the initial query representations using a query network $f_q$ implemented as a two-layer feed-forward neural network:

$$\mathbf{Q}^{(0)} = \sigma \left( \left[ \sigma(\mathbf{S}\mathbf{W}_0) \parallel \mathbf{X} \right] \mathbf{W}_1 \right) \tag{6}$$

---

[1]We use initial node representation and initial query representation interchangeably throughout the paper.

where $\mathbf{W}_0 \in \mathbb{R}^{n \times d_{in}}$ and $\mathbf{W}_1 \in \mathbb{R}^{2d_{in} \times d_0}$ are the parameters of the query networks, $\|$ is the concatenation operator, and $\sigma$ is the non-linearity implemented using LeakyReLU.

### 3.2.1 PERMUTATION INVARIANCE

Considering the inherent permutation-invariant property of graph-structured data, a model designed to address graph classification tasks, should also enforce this property. This implies that the model should generate same outputs for isomorphic input graphs. We impose this on GMN architecture by sorting the topological embedding row-wise as a pre-processing step.

**Proposition 1.** *Given a sorting function o,* $\mathrm{GMN}\left(o\left(\mathbf{S}\right), \mathbf{X}\right)$ *is permutation-invariant.*

*Proof.* Let $\mathbf{P} \in \{0, 1\}^{\{n \times n\}}$ be an arbitrary permutation matrix. For each node in graph $\mathbf{G}$ with adjacency matrix $\mathbf{A}$, the corresponding node in graph $\mathbf{G}_P$ with permuted adjacency matrix $\mathbf{PAP}^T$ has the permuted version of the topological embedding of the node in graph $\mathbf{G}$. Sorting the embeddings cancels out the effect of permutation and makes the corresponding embeddings in graph $\mathbf{G}$ and $\mathbf{G}_P$ identical.

### 3.3 MEMGNN ARCHITECTURE

Unlike the GMN architecture, the query network in MemGNN relies on message passing to compute the initial query $\mathbf{Q}_0$ as follows:

$$\mathbf{Q}^{(0)} = G_\theta\left(\mathbf{A}, \mathbf{X}\right) \tag{7}$$

where query network $G_\theta$ is an arbitrary parameterized message passing GNN (Gilmer et al., 2017; Li et al., 2015; Kipf & Welling, 2016; Velikovi et al., 2018).

In our implementation, we use a modified variant of GAT (Velikovi et al., 2018). Specifically, we introduce an extension to the original GAT model called edge-based GAT (**e-GAT**) and use it as the query network. Unlike GAT, e-GAT learns attention weights not only from the neighbor nodes but also from the input edge features. This is especially important for data containing edge information (e.g., various bonds among atoms represented as edges in molecule datasets). In an e-GAT layer, attention score between two neighbor nodes is computed as follows:

$$\alpha_{ij} = \frac{\exp\left(\sigma\left(\mathbf{W}\left[\mathbf{W}_n h_i^{(l)} \| \mathbf{W}_n h_j^{(l)} \| \mathbf{W}_e h_{i \to j}^{(l)}\right]\right)\right)}{\sum\limits_{k \in \mathcal{N}_i} \exp\left(\sigma\left(\mathbf{W}\left[\mathbf{W}_n h_i^{(l)} \| \mathbf{W}_n h_k^{(l)} \| \mathbf{W}_e h_{i \to k}^{(l)}\right]\right)\right)} \tag{8}$$

where $h_i^{(l)}$ and $h_{i \to j}^{(l)}$ denote the representation of node $i$ and the representation of the edge connecting node $i$ to its one-hop neighbor node $j$ in layer $l$, respectively. $\mathbf{W}_n$ and $\mathbf{W}_e$ are learnable node and edge weights and $\mathbf{W}$ is the parameter of a single-layer feed-forward network that computes the attention score. $\sigma$ is the non-linearity implemented using LeakyReLU.

### 3.4 TRAINING

We jointly train the model using two loss functions: a supervised loss and an unsupervised clustering loss. The supervised loss denoted as $\mathcal{L}_{sup}$ is defined as cross-entropy loss and root mean square error (RMSE) for graph classification and regression tasks, respectively. The unsupervised clustering loss is inspired by deep clustering methods (Razavi et al., 2019; Xie et al., 2016; Aljalbout et al., 2018). It encourages the model to learn clustering-friendly embeddings in the latent space by learning from high confidence assignments with the help of an auxiliary target distribution. The unsupervised loss is defined as the Kullback-Leibler (KL) divergence between the soft assignments $\mathbf{C}^{(l)}$ and the auxiliary distribution $\mathbf{P}^{(l)}$ as follows:

$$\mathcal{L}_{KL}^{(l)} = \text{KL}\left(\mathbf{P}^{(l)}||\mathbf{C}^{(l)}\right) = \sum_i \sum_j P_{ij}^{(l)} log \frac{P_{ij}^{(l)}}{C_{ij}^{(l)}} \tag{9}$$

For the target distributions $\mathbf{P}^{(l)}$, we use the distribution proposed in (Xie et al., 2016) which normalizes the loss contributions and improves the cluster purity while emphasizing on the samples with higher confidence. This distribution is defined as follows:

$$P_{ij}^{(l)} = \frac{\left(C_{ij}^{(l)}\right)^2 / \sum\limits_i C_{ij}^{(l)}}{\sum\limits_{j'} \left(C_{ij'}^{(l)}\right)^2 / \sum\limits_i C_{ij'}^{(l)}} \tag{10}$$

We define the total loss as follows where $L$ is the number of memory layers and $\lambda \in [0, 1]$ is a scalar weight.

$$\mathcal{L} = \frac{1}{N} \sum_{n=1}^{N} \left(\lambda \mathcal{L}_{sup} + (1 - \lambda) \sum_{l=1}^{L} \mathcal{L}_{KL}^{(l)}\right) \tag{11}$$

We initialize the model parameters, the keys, and the queries randomly and optimize them jointly with respect to $\mathcal{L}$ using mini-batch stochastic gradient descent. To stabilize the training, the gradients of $\mathcal{L}_{sup}$ are back-propagated batch-wise while the gradients of $\mathcal{L}_{KL}^{(l)}$ are applied epoch-wise by periodically switching $\lambda$ between 0 and 1. Updating the centroids, i.e., memory keys, with the same frequency as the network parameters can destabilize the training. To address this, we optimize all model parameters and the queries in each batch with respect to $\mathcal{L}_{sup}$ and in each epoch with respect to $\mathcal{L}_{KL}^{(l)}$. Memory keys, on the other hand, are only updated at the end of each epoch by the gradients of $\mathcal{L}_{KL}^{(l)}$. This technique has also been applied in (Hassani & Haley, 2019; Caron et al., 2018) to avoid trivial solutions.

## 4 EXPERIMENTS

### 4.1 DATASETS

We use nine benchmarks including seven graph classification and two graph regression datasets to evaluate the proposed method. These datasets are commonly used in both graph kernel (Borgwardt & Kriegel, 2005; Yanardag & Vishwanathan, 2015; Shervashidze et al., 2009; Ying et al., 2018; Shervashidze et al., 2011; Kriege et al., 2016) and GNN (Cangea et al., 2018; Ying et al., 2018; Lee et al., 2019; Gao & Ji, 2019) literature. The summary of the datasets is as follows where the first two benchmarks are regression tasks and the rest are classification tasks.

**ESOL** (Delaney, 2004) contains water solubility data for compounds.
**Lipophilicity** (Gaulton et al., 2016) contains experimental results of octanol/water distribution of compounds.
**Bace** (Subramanian et al., 2016) provides quantitative binding results for a set of inhibitors of human $\beta$-secretase 1 (BACE-1).
**DD** (Dobson & Doig, 2003) is used to distinguish enzyme structures from non-enzymes.
**Enzymes** (Schomburg et al., 2004) is for predicting functional classes of enzymes.
**Proteins** (Dobson & Doig, 2003) is used to predict the protein function from structure.
**Collab** (Yanardag & Vishwanathan, 2015) is for predicting the field of a researcher given one's ego-collaboration graph.
**Reddit-Binary** (Yanardag & Vishwanathan, 2015) is for predicting the type of community given a graph of online discussion threads.
**Tox21** (Challenge, 2014) is for predicting toxicity on 12 different targets.

For more information about the detailed statistics of the datasets refer to Appendix A.2.

Table 1: Mean validation accuracy over 10-folds.

| | Method | Dataset | | | | |
|---|---|---|---|---|---|---|
| | | Enzymes | Proteins | DD | Collab | Reddit-B |
| Kernel | Graphlet (Shervashidze et al., 2009) | 41.03 | 72.91 | 64.66 | 64.66 | 78.04 |
| | ShortestPath (Borgwardt & Kriegel, 2005) | 42.32 | 76.43 | 78.86 | 59.10 | 64.11 |
| | WL (Shervashidze et al., 2011) | 53.43 | 73.76 | 74.02 | 78.61 | 68.20 |
| | WL Optimal (Kriege et al., 2016) | 60.13 | 75.26 | 79.04 | **80.74** | 89.30 |
| GNN | PatchySan (Niepert et al., 2016) | – | 75.00 | 76.27 | 72.60 | 86.30 |
| | GraphSage (Hamilton et al., 2017) | 54.25 | 70.48 | 75.42 | 68.25 | – |
| | ECC (Simonovsky & Komodakis, 2017) | 53.50 | 72.65 | 74.10 | 67.79 | – |
| | Set2Set (Vinyals et al., 2015) | 60.15 | 74.29 | 78.12 | 71.75 | – |
| | SortPool (Morris et al., 2019) | 57.12 | 75.54 | 79.37 | 73.76 | – |
| | DiffPool (Ying et al., 2018) | 60.53 | 76.25 | 80.64 | 75.48 | 85.95 |
| | CliquePool (Luzhnica et al., 2019) | 60.71 | 72.59 | 77.33 | 74.50 | – |
| | Sparse HGC (Cangea et al., 2018) | 64.17 | 75.46 | 78.59 | 75.46 | 79.20 |
| | TopKPool (Gao & Ji, 2019) | – | 77.68 | 82.43 | 77.56 | 74.70 |
| | SAGPool (Lee et al., 2019) | – | 71.86 | 76.45 | – | 73.90 |
| | GMN (ours) | **78.66** | **82.25** | **84.40** | 80.18 | **95.28** |
| | MemGNN (ours) | 75.50 | 81.35 | 82.92 | 77.0 | 85.55 |

## 4.2 GRAPH CLASSIFICATION RESULTS

To evaluate the performance of our models on DD, Enzymes, Proteins, Collab, and Reddit-Binary datasets, we follow the experimental protocol in (Ying et al., 2018) and perform 10-fold cross-validation and report the mean accuracy over all folds. We also report the performance of four graph kernel methods including Graphlet (Shervashidze et al., 2009), shortest path (Borgwardt & Kriegel, 2005), Weisfeiler-Lehman (WL) (Shervashidze et al., 2011), and WL Optimal Assignment (Kriege et al., 2016), and ten GNN models.

The results shown in Table 1 suggest that: (*i*) our models achieve state-of-the-art results w.r.t. GNN models and significantly improve the performance on Enzymes, Proteins, DD, Collab, and Reddit-Binary datasets by absolute margins of 14.49%, 6.0%, 3.76%, 2.62%, and 8.98% accuracy, respectively, (*ii*) our models outperform graph kernels on all datasets except Collab where our models are competitive with the best kernel, i.e., absolute margin of 0.56%, (*iii*) both proposed models achieve better performance or are competitive compared to the baseline GNNs, (*iv*) GMN achieves better results compared to MemGNN which suggests that replacing local adjacency information with global topological embeddings provides the model with more useful information, and (*v*) On Collab, our models are outperformed by a variant of DiffPpool (i.e., diffpool-det) (Ying et al., 2018) and WL Optimal Assignment (Kriege et al., 2016). The former is a GNN augmented with deterministic clustering algorithm[2], whereas the latter is a graph kernel method. We speculate this is because of the high edge-to-node ratio in this dataset and the augmentations used in these two methods help them with extracting near-optimal cliques.

To evaluate the performance on the BACE and Tox21 datasets, we follow the evaluation protocol in (Wu et al., 2018) and report the area under the curve receiver operating characteristics (AUC-ROC) measure. Considering that the BACE and Tox21 datasets contain initial edge features, we train the MemGNN model and compare its performance to the baselines reported in (Wu et al., 2018). The results shown in Table 2 suggest that our model achieves state-of-the-art results by absolute margin of 4.0 AUC-ROC on the BACE benchmark and is competitive with the state-of-the-art GCN model on the Tox21 dataset, i.e., absolute margin of 0.001.

## 4.3 GRAPH REGRESSION RESULTS

For the ESOL and Lipophilicity datasets, we follow the evaluation protocol in (Wu et al., 2018) and report their RMSEs. Considering that these datasets contain initial edge features (refer to Appendix A.2 for further details), we train the MemGNN model and compare the results to the baseline mod-

---

[2]In diffpool-det assignment matrices are generated using a deterministic graph clustering algorithm.

Table 2: AUC-ROC on BACE and Tox21 datasets.

| Method | Dataset | | | |
|---|---|---|---|---|
| | BACE | | Tox21 | |
| | validation | test | validation | test |
| Logistic Regression | $0.719 \pm 0.003$ | $0.781 \pm 0.010$ | $0.772 \pm 0.011$ | $0.794 \pm 0.015$ |
| KernelSVM | $0.739 \pm 0.000$ | $0.862 \pm 0.000$ | $0.818 \pm 0.010$ | $0.822 \pm 0.006$ |
| XGBoost | $0.756 \pm 0.000$ | $0.850 \pm 0.008$ | $0.775 \pm 0.018$ | $0.794 \pm 0.014$ |
| Random Forest | $0.728 \pm 0.004$ | $0.867 \pm 0.008$ | $0.763 \pm 0.002$ | $0.769 \pm 0.015$ |
| IRV | $0.715 \pm 0.001$ | $0.838 \pm 0.000$ | $0.807 \pm 0.006$ | $0.799 \pm 0.006$ |
| Multitask | $0.696 \pm 0.037$ | $0.824 \pm 0.0006$ | $0.795 \pm 0.017$ | $0.803 \pm 0.012$ |
| Bypass | $0.745 \pm 0.017$ | $0.829 \pm 0.006$ | $0.800 \pm 0.008$ | $0.810 \pm 0.013$ |
| GCN | $0.627 \pm 0.015$ | $0.783 \pm 0.014$ | $0.825 \pm 0.013$ | $\mathbf{0.829 \pm 0.006}$ |
| Weave | $0.638 \pm 0.014$ | $0.806 \pm 0.002$ | $0.828 \pm 0.008$ | $0.820 \pm 0.010$ |
| MemGNN (ours) | $\mathbf{0.859 \pm 0.000}$ | $\mathbf{0.907 \pm 0.000}$ | $\mathbf{0.862 \pm 0.009}$ | $0.828 \pm 0.004$ |

Table 3: RMSE on ESOL and Lipophilicity datasets.

| Method | Dataset | | | |
|---|---|---|---|---|
| | ESOL | | Lipophilicity | |
| | validation | test | validation | test |
| Multitask | $1.17 \pm 0.13$ | $1.12 \pm 0.19$ | $0.852 \pm 0.048$ | $0.859 \pm 0.013$ |
| Random Forest | $1.16 \pm 0.15$ | $1.07 \pm 0.19$ | $0.835 \pm 0.036$ | $0.876 \pm 0.040$ |
| XGBoost | $1.05 \pm 0.10$ | $0.99 \pm 0.14$ | $0.783 \pm 0.021$ | $0.799 \pm 0.054$ |
| GCN | $1.05 \pm 0.15$ | $0.97 \pm 0.01$ | $0.678 \pm 0.040$ | $0.655 \pm 0.036$ |
| MPNN | $0.55 \pm 0.02$ | $0.58 \pm 0.03$ | $0.757 \pm 0.030$ | $0.715 \pm 0.035$ |
| KRR | $1.65 \pm 0.19$ | $1.53 \pm 0.06$ | $0.889 \pm 0.009$ | $0.899 \pm 0.043$ |
| DAG | $0.74 \pm 0.04$ | $0.82 \pm 0.08$ | $0.857 \pm 0.050$ | $0.835 \pm 0.039$ |
| Weave | $0.57 \pm 0.04$ | $0.61 \pm 0.07$ | $0.734 \pm 0.011$ | $0.715 \pm 0.035$ |
| MemGNN (ours) | $\mathbf{0.53 \pm 0.03}$ | $\mathbf{0.54 \pm 0.01}$ | $\mathbf{0.555 \pm 0.039}$ | $\mathbf{0.556 \pm 0.023}$ |

els reported in (Wu et al., 2018) including graph based methods such as GCN, MPNN, directed acyclic graph (DAG) model, and Weave as well as other conventional methods such as kernel ridge regression (KRR) and influence relevance voting (IRV). Results shown in Table 3 suggest that our MemGNN model achieves state-of-the-art results by absolute margin of 0.07 and 0.1 RMSE on ESOL and Lipophilicity benchmarks, respectively. For further details on regression datasets and baselines please refer to (Wu et al., 2018).

## 4.4 ABLATION STUDY

### 4.4.1 EFFECT OF EDGE FEATURES

To investigate the effect of the proposed e-GAT model, we train the MemGNN model using both GAT and e-GAT layers as the query network. Considering that the ESOL, Lipophilicity, and BACE datasets contain edge features, we use them as the benchmarks. Since nodes have richer features compared to edges, we set the node and edge feature dimensions to 16 and 4, respectively. The performance of the two layers on the ESOL dataset shown in AppendixA.3 suggesting that e-GAT achieves better results on the validation set in each epoch compared to the standard GAT model. We observed the same effect on Lipophilicity and BACE datasets.

### 4.4.2 EFFECT OF TOPOLOGICAL EMBEDDING

To investigate the effect of topological embeddings on the GMN model, we evaluated three initial topological features including adjacency matrix, normalized adjacency matrix, and RWR. For further details on RWR, see section A.4. The results suggest that using the RWR as the initial positional embedding achieves the best performance. For instance, 10-fold cross validation accuracy of a GMN model trained on Enzymes dataset with adjacency matrix, normalized adjacency matrix, and RWR are 78.66%, 77.16%, and 77.33%, respectively. Furthermore, sorting the topological embeddings

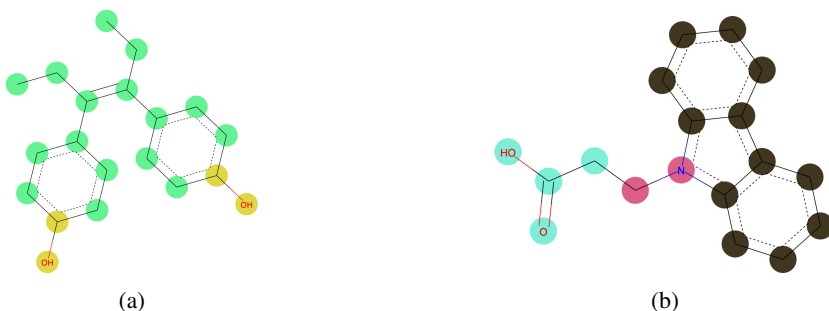

(a)          (b)

Figure 2: Visualization of the learned clusters of two molecule instances from (a) ESOL and (b) Lipophilicity datasets. The visualizations show that the learned clusters correspond to known chemical groups. Note that a node without label represents a carbon atom. For more visualizations and discussion see section A.5

to guarantee invariance to permutations improves the performance. For example, it increases the accuracy on the DD dataset from 82.24% to 84.40%.

### 4.4.3 DOWN-SAMPLING NEIGHBORS WITH RANDOM WALKS

We investigate two methods to down-sample the neighbors in dense datasets such as Collab (i.e., average of 66 neighbors per node) to enhance the memory and computation. The first method randomly selects 10% of the edges whereas the second method ranks the neighbors based on their RWR scores with respect to the center node and then keeps the top 10% of the edges. We trained the MemGNN model on Collab using both sampling methods which resulted in 73.9% and 73.1% 10-fold cross validation accuracy for random and RWR-based sampling methods respectively, suggesting that random sampling performs slightly better than RWR based sampling.

### 4.4.4 EFFECT OF NUMBER OF KEYS AND HEADS

We speculate that although keys represent the clusters, the number of keys is not necessarily proportional to the number of the nodes in the input graphs. In fact, datasets with smaller graphs might have more meaningful clusters to capture. For example, molecules are comprised of many functional groups and yet the average number of nodes in the ESOL dataset is 13.3. Moreover, our experiments show that for Enzymes with average number of 32.69 nodes, the best performance is achieved with 10 keys whereas for the ESOL dataset 64 keys results in the best performance. In ESOL 8, 64, and 160 keys result in RMSE of 0.56, 0.52, and 0.54, respectively. We also observed that with a fixed number of parameters, increasing the number of memory heads improves the performance. For instance, when the model is trained on ESOL with 160 keys and 1 head, it achieves RMSE of 0.54, whereas when trained with 32 keys and 5 heads, the same model achieves RMSE of 0.53.

### 4.4.5 WHAT DO THE KEYS REPRESENT?

Intuitively, the memory keys represent the cluster centroids and enhance the model performance by capturing meaningful structures. To investigate this, we used the learned keys to interpret the knowledge learned by models through visualizations. Figure 2 visualizes the learned clusters over atoms (i.e., atoms with the same color are within the same cluster) indicating that the clusters mainly consist of meaningful chemical substructures such as a carbon chain and a Hydroxyl group (OH) (i.e., Figure 2a), as well as a Carboxyl group (COOH) and a benzene ring (i.e., Figure 2b). From a chemical perspective, Hydroxyl and Carboxyl groups, and carbon chains have a significant impact on the solubility of the molecule in water or lipid. This confirms that the network has learned chemical features that are essential for determining the molecule solubility. It is noteworthy that we tried initializing the memory keys using K-Means algorithm over the initial node representations to warm-start them but did not observe any significant improvement over the randomly selected keys.

## 5    CONCLUSION

We proposed an efficient memory layer and two models for hierarchical graph representation learning. We evaluated the proposed models on nine graph classification and regression tasks and achieved state-of-the-art results on eight of them. We also experimentally showed that the learned representations can capture well-known chemical features of the molecules. Furthermore, we showed that concatenating node features with topological embeddings and passing them through a few memory layers achieves notable results without using message passing. Moreover, we showed that defining the topological embeddings using graph diffusion achieves best performance. Finally, we showed that although connectivity information is not explicitly imposed on the model, the memory layer can process node representations and properly cluster and aggregate the learned representations.

**Limitations**: In section 4.3, we discussed that a graph kernel and a GNN augmented with deterministic clustering achieve better performance compared to our models on the Collab dataset. Analyzing samples in this dataset suggests that in graphs with dense communities, such as cliques, our model struggles to properly detect the dense sub-graphs.

**Future Directions**: We plan to extend our models to also perform node classification by attending to the node representations and centroids of the clusters from different layers of hierarchy that the nodes belongs to. Moreover, we are planning to evaluate other graph diffusion models, e.g., personalized PageRank and heat kernel, to initialize the topological embeddings. We are also planing to investigate the representation learning capabilities of the proposed models in self-supervised setting.

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

## A    APPENDIX

### A.1    IMPLEMENTATION DETAILS

We implemented the model with PyTorch (Paszke et al., 2017) and optimized it using Adam (Kingma & Ba, 2014) optimizer. We trained the model for a maximum number of 2000 epochs and decayed the learning rate every 500 epochs by 0.5. The model uses batch-normalization (Ioffe & Szegedy, 2015), skip-connections, LeakyRelu activation functions, and dropout (Srivastava et al., 2014) for regularization. We also set the temperature in Students t-distribution to 1.0 and the restart probability in RWR to 0.1. We decided the hidden dimension and the number of model parameters using random hyper-parameter search strategy. The best performing hyper-parameters for the datasets are shown in Table 4.

### A.2    DATASET STATISTICS

Table 5 summarizes the statistics of the datasets used for graph classification and regression tasks.

### A.3    EFFECT OF E-GAT

In section 4.4.1, we introduced e-GAT. Figures 3a and 3b illustrate the RMSE and $R^2$ score on the validation set of the ESOL dataset achieved by a MemGNN model using both GAT and e-GAT as the query network, respectively. As shown, e-GAT performs better compared to GAT on both metrics.

### A.4    RANDOM WALK WITH RESTART

Suppose an agent randomly traverses a graph starting from node $i$ and iteratively walks towards its neighbors with a probability proportional to the edge weight that connects them. The agent also can randomly restart the traverse with probability $p$. Eventually, the agent will stop at node $j$ with a probability called relevance score of node $j$ with respect to node $i$ (Tong et al., 2006). The relevance score of node $i$ with every other node of the graph is defined as follows:

$$\vec{t_i} = p\tilde{\mathbf{A}}\vec{t_i} + (1-p)\vec{e_i} = (1-p)(I - p\tilde{\mathbf{A}})^{-1}\vec{e_i} \tag{12}$$

where $\vec{t_i}$ is the RWR score corresponding to node $i$, $p$ is the restart probability, $\tilde{\mathbf{A}}$ is the normalized adjacency matrix, and $\vec{e_i}$ is one-hot vector representation of node $i$.

Note that the restart probability defines how far the agent can walk from the source node and therefore $\vec{t_i}$ represents the trad-off between local and global information around node $i$.

### A.5    LEARNED CLUSTERS

Figure 4 shows how unsupervised loss helps the model to push the nodes into distinct clusters. Figures 4a and 4c illustrates clusters with unsupervised loss and Figures 4b and 4d show computed clusters without unsupervised loss. The visualizations suggest that unsupervised loss helps the model to avoid trivial solutions by collapsing the latent node representations into meaningless clusters. Also, Figure 5 represents meaningful chemical groups extracted by MemGNN. Figures 5b and 5d are from LIPO and figure 5a and 5c are from ESOL dataset respectively.

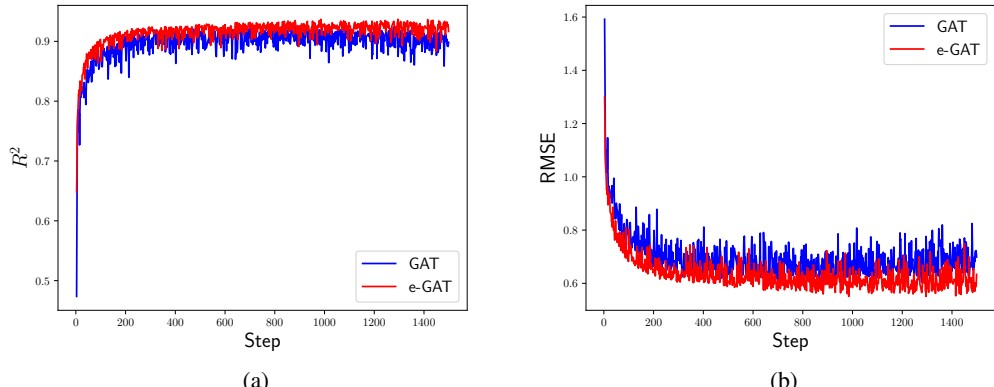

Figure 3: Validation (a) $R^2$ score, and (b) RMSE achieved by MemGNN model on ESOL with GAT and e-GAT based query networks.

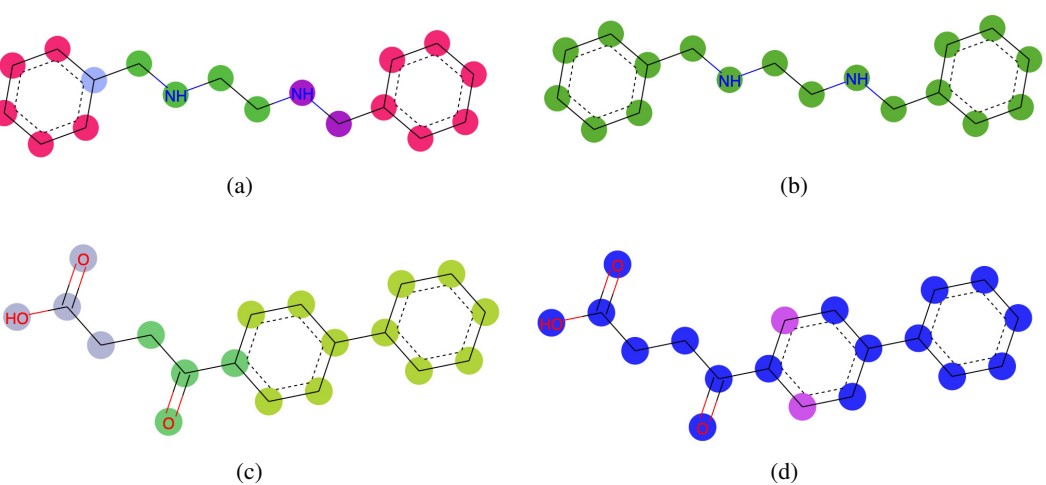

Figure 4: Figures (b) and (d) show computed clusters without using unsupervised clustering loss, whereas Figures (a) and (c) show the clusters learned using the unsupervised clustering loss. The visualizations suggest that the unsupervised loss helps the model in learning distinct and meaningful clusters.

Table 4: Hyper-parameters selected for the models.

| Dataset | #Keys | #Heads | #Layers | Hidden Dimension | Batch |
|---------|-------|--------|---------|------------------|-------|
| Enzymes | [10,1] | 5 | 2 | 100 | 20 |
| Proteins | [10,1] | 5 | 2 | 80 | 20 |
| DD | [16, 8, 1] | 5 | 3 | 120 | 64 |
| Collab | [32, 8, 1] | 5 | 3 | 100 | 64 |
| Reddit-B | [32,1] | 1 | 2 | 16 | 32 |
| ESOL | [64,1] | 5 | 2 | 16 | 32 |
| Lipophilicity | [32,1] | 5 | 2 | 16 | 32 |
| BACE | [32,1] | 5 | 2 | 8 | 32 |

Table 5: Summary of statistics of the benchmark dataset.

| Name | Task | Graphs | Classes | Avg. Nodes | Avg. Edges | Node Attr. | Edge Attr. |
|------|------|--------|---------|------------|------------|------------|------------|
| Enzymes | classification | 600 | 6 | 32.63 | 62.14 | 18 | 0 |
| Proteins | classification | 1113 | 2 | 39.06 | 72.82 | 29 | 0 |
| DD | classification | 1178 | 2 | 284.32 | 715.66 | 0 | 0 |
| Collab | classification | 5000 | 3 | 74.49 | 2475.78 | 0 | 0 |
| Reddit-B | classification | 2000 | 2 | 429.63 | 497.75 | 0 | 0 |
| Bace | classification | 1513 | 2 | 34.09 | 36.86 | 32 | 7 |
| Tox21 | classification | 8014 | 2 | 17.87 | 18.50 | 32 | 7 |
| ESOL | regression | 1144 | - | 13.29 | 13.68 | 32 | 7 |
| Lipophilicity | regression | 4200 | - | 27.04 | 29.50 | 32 | 7 |

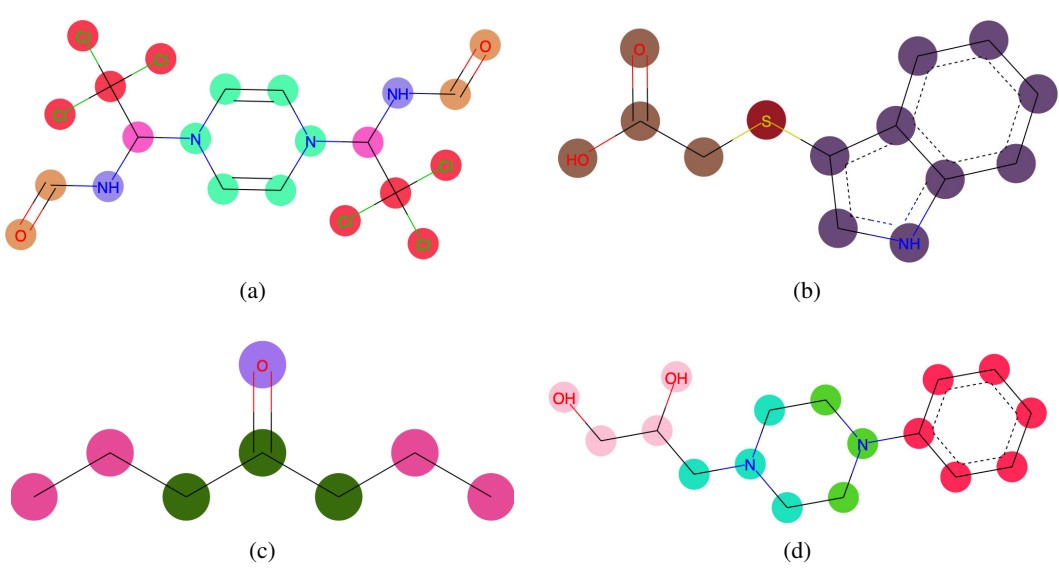

(a)        (b)

(c)        (d)

Figure 5: Clusters learned by a MeMGNN for ESOL and LIPO dataset. Chemical groups like OH (hydroxyl group), CCl3, COOH (carboxyl group), CO (ketone group) as well as benzene rings have been recognized during the learning procedure. These chemical groups are highly active and have a great impact on the solubility of molecules.

