# OpenReview forum: "Memory-Based Graph Networks"
_ICLR.cc/2020/Conference — Accept (Poster)_

### Official Review · AnonReviewer3 · 2019-10-23
**Official Blind Review #3**

**Rating:** 6

**Review:**

The paper presents "memory layer" to simultaneously do graph representation learning and pooling in a hierarchical way. It shares the same spirit with the previous models (DiffPool and Mincut pooling) which cluster nodes and learn representation of the coarsened graph. In DiffPool, Graph convolutional Neural Networks (GCNs) with 2-6 iterations of “Message passing” is used to learn node embedding, followed by graph pooling. By contrast, the proposed model circumvent the inefficiency of using message passing by using their proposed memory layer.

Pros:
* The paper is generally written well, but some important details are missing.
* Clear visualization of the results.
* Useful ablation study.

Cons:
* Missing information, which can be critical for the success of the model:
(a) the estimation of the keys, and
(b) how does the convolutional layer in Eq (2) work, given that the input for it is concatenation of matrices, which has no spatial structure?
* Experiments on graph classification lack diversity, where CoLLAB is the only non-chemical dataset in the experiment.
* The paper argues that interactive message passing is not efficient. But do you have any explanation on why MemGNN with message passing in initial embedding learning performs better than GMN without message passing in D&D dataset?
* I have some reservation for calling something "memory", which is meant to store information for later processing. For this work, the network is a feed-forward architecture for processing graphs, where the middle layers (the queries) are matrices, which can be studied on their own right (e.g., see [1]).

At this point, the ideas for graph representation are plentiful, but there have not been a coherent story on how and why new architectures should work better than previous ones. This paper can be made stronger by offering insights along this line.

[1] Do, K., Tran, T., & Venkatesh, S. (2017). Learning deep matrix representations. arXiv preprint arXiv:1703.01454.




**Experience Assessment:**

I have published in this field for several years.

**Review Assessment: Checking Correctness Of Derivations And Theory:**

I assessed the sensibility of the derivations and theory.

**Review Assessment: Checking Correctness Of Experiments:**

I assessed the sensibility of the experiments.

**Review Assessment: Thoroughness In Paper Reading:**

I read the paper at least twice and used my best judgement in assessing the paper.

---

> ### Author Response · Authors · 2019-11-13
> **Response to Review #3 - part 2**
>
> - We tried to address this in Section 1. Specifically, the initial GNNs were mostly based on RNNs that suffered from high computational overhead. The GCNs came next and were more successful. However, they use: (1) structure-dependent aggregation, and (2) use a predetermined normalization constant (e.g., it is proportional to node degrees in original GCN model). Attention-based GNNs on the other hand learn the contribution of neighbors and achieve best performance on node representation learning. These models, however, are inherently flat and hence do not exploit the hierarchical structures within the graphs. And because of this reason they perform poorly in graph classification task in which they use simple arithmetic pooling to directly compute the graph embedding from all node embeddings. To address this, recent work introduced end-to-end pooling layers by defining the pooling layer as another neural network. All this work places focuses only on local information in which information is propagated via message passing. We, on the other hand, let the attention mechanism decide each node should attend to which nodes and we do not constraint the node selection to any explicit neighborhood definition. We speculate this results in stronger graph embedding compared to previous work (i.e., similar effect is observed when comparing Transformers to RNNs such as LSTMs). Please note that a good review paper explaining the evolution of GNNs can be found in [10].
>
>
> References
> [1] Felix Hill et al. The Goldilocks Principle: Reading Children's Books with Explicit Memory Representations. arXiv 2015.
> [2] Kaveh Hassani et al. Unsupervised multi-task feature learning on point clouds. ICCV 2019.
> [3] Junyuan Xie et al. Unsupervised deep embedding for clustering analysis. ICML 2016.
> [4] Mathilde Caron et al. Deep Clustering for Unsupervised Learning of Visual Features. ECCV 2018.
> [5] Alex Graves et al. Neural turing machines. arXiv 2014.
> [6] Jason Weston  et al. Memory networks. ICLR 2015.
> [7] Jack Rae et al. Scaling memory-augmented neural networks with sparse reads and writes. NeurIPS 2016.
> [8] Guillaume Lample et al. Large Memory Layers with Product Keys. arXiv 2019.
> [9] Kien Do et al. Learning deep matrix representations. arXiv 2017.
> [10] Zonghan Wu et al. A Comprehensive Survey on Graph Neural Networks. arXiv 2019.
> [11] Paul D. Dobson et al. Distinguishing Enzyme Structures from Non-enzymes Without Alignments. J. Mol. Biol. .2003

---

> ### Author Response · Authors · 2019-11-13
> **Response to Review #3 - part 1**
>
> We thank the reviewer for the insightful review and comments on our work! We address the questions and comments as follows:
>
> - We revised section 3 to add the requested details. In short: the keys are initialized randomly and then are updated epoch-wise w.r.t the unsupervised loss (i.e., KL divergence). For the initialization, we also tried preloading the memory (i.e., similar to [1]) using centroids computed by K-Means over the initial node embeddings to warm-start them but we did not observe any significant improvement over the randomly selected keys. We optimize all model parameters except the keys in each batch w.r.t Eq. (12). That means parameters are updated by the gradients of the cross-entropy loss in each batch and also are updated by the gradient of the KL divergence at the end of each epoch. Keys, on the other hand, are only updated at the end of each epoch by the gradient of the KL divergence. Updating the centroids with the same frequency as the network parameters can destabilize the training [2, 3, 4]. This is why we update them epoch-wise.
>
> -  We stack the matrices to form a tensor of size $h \times n_l \times n_{l+1}$ where $h$ is the number of heads (i.e., depth in standard convolution analogy), and $n_l \times n_{l+1}$ is the size of the cluster assignment matrix $C$ (i.e., height and width in standard convolution analogy). In other words, we treat each head as a separate channel. As you mentioned, because there is no spatial structure, we use $[1 \times 1]$ convolution to aggregate the $C_{j,j}$s across channels and therefore the convolution behaves as a weighted pooling that reduces the heads to one head. We then pass the aggregated matrix to a softmax function that is applied in a row-wise fashion. We updated the paper to address this by changing Eq. (2) and explaining the convolution in more details.
>
> -  We added the results on Reddit-binary graph classification benchmark (i.e., predicting community type in Reddit discussions) to the revised paper. The GMN model archives state-of-the-art accuracy of 86.39% on this dataset. Per request from Reviewer #2, we also added the results on Tox21 graph classification benchmark. The results of these two datasets are reported in Appendix 3, Tables 6 and 7, respectively.
>
> - We speculate the DD dataset is unique among the studied benchmarks as it relies more on local information. This is explicitly shown in [11] where the authors conclude that the most important features to train an SVM on this dataset are surface features which have local behavior. This is why the GMN model which pays more attention to global information is outperformed by our MemGNN model which captures local interactions through message passing. We added this to the revised paper (conclusion section).
>
> - We agree that in work such as Neural Turing Machines [5], Memory Networks [6], and Sparse Access Memory [7],  a memory component is explicitly defined as a decoupled unit from the model parameters with soft read/write access. However, in most recent work such as Key-Value Memory Layers [8], the memory layer is treated as another layer interleaved and stacked with other standard layers such as self-attention or FFN layers. Following this line of inquiry, we also call the layer as a memory layer because although it is a feed-forward layer but it introduces key-value like memory to the network.
> Regarding [9], we can also formulate our memory layer with matrix representation as it is explicitly mentioned in the paper: “Indeed, several memory augmented networks such as Neural Turing Machine …. can be formulated in similar ways with P being the external memory and U is the collection of read heads.” This work is suggesting a general improvement to deep models (feed-forward, attention, memory, recurrent, etc.) by using matrix representation instead of vector representation. We, on the other hand, are specifically introducing a memory layer that can be stacked with other standard layers and emulates a key-value memory. That said, we welcome any naming suggestions for this layer.

---

### Official Review · AnonReviewer1 · 2019-10-25
**Official Blind Review #1**

**Rating:** 6

**Review:**

The paper proposes to use memory network (self-attention) to summarize information from graphs. The proposed approach "soft clusters" the node embeddings into coarser representation and eventually fed to a final MLP for classification. The empirical results on some standard datasets show promising gains of the algorithm.

The proposed approach stacks a few layers of self-attention on top of either some features of the nodes (including projected edge connections, parametrized by W_0) or the node embeddings of some form of graph neural network. And this stacking seems to be simple combination of existing approaches, without fully integrating them. In fact, the training process is also separated: "task-specific loss are back-propagated batch-wise while the gradients of the unsupervised loss are applied
epoch-wise". It makes me wonder whether we can just separate it into two stages, i.e., first learn a node embedding using graph neural network, then learn this self attention transformation. Due to the above issues, I feel the novelty of the approach is limited and incremental.

Another issue with the paper is that the notations seem to be messed up and some concepts are not explained clearly. For example, the C_{i,j} soft assignment matrix is normalized row-wise, then Eqn (3) seems very suspicious, because it averages the queries using weights along the other direction, thus not normalized. Also, the dimension of the MLP weights do not align well with inputs, for instance in Eqn (4), it should be written as V^(l) W.

There are more questions that are not clearly specified in the current manuscript. For example, where does the keys K come from? From the text description, it seems to be cluster results, and do you do the clustering on every gradient update? Or are they learned from scratch? The distribution P defined in Eqn (11) also seems to be difficult to optimize since it depends on C_{i,j} and is connected to different entries. Is simple SGD sufficient to optimize over P? Moreover, in the experiment section, it is unknown how many layers of self-attention is applied and what are the important parameters. For better comparison, the experiment section should include some estimate of parameter size as well.

The experiment results seem interesting since the approach indeed achieves good performance across many datasets. Also the visualized keys are interesting as well because it captures some meaningful patterns from the data.

There is some related work that you should cite:
Hanjun Dai, Bo Dai and Le Song. Discriminateive Embeddings of Latent Variable Models for Structured Data. International Conference on Machine Learning (ICML) 2016.


============================================================
Based on the authors' reply and other reviews, I have changed my rating to "Weak accept".

The authors' reply has clarified the important detail of how the key matrix is learned and now it is clear that the algorithm is not just two-stage separate learning. In light of this clarification, I think the proposed algorithm is novel enough and the jointly training mechanism is also beneficial for the state-of-the-arts results reported in the experiments.

**Experience Assessment:**

I have read many papers in this area.

**Review Assessment: Checking Correctness Of Derivations And Theory:**

I assessed the sensibility of the derivations and theory.

**Review Assessment: Checking Correctness Of Experiments:**

I assessed the sensibility of the experiments.

**Review Assessment: Thoroughness In Paper Reading:**

I read the paper at least twice and used my best judgement in assessing the paper.

---

> ### Author Response · Authors · 2019-11-13
> **Response to Review #1**
>
> We would like to thank the reviewer for insightful and detailed comments. Our responses are as follows:
>
> -  To be clear, there is a difference between decoupling the training and training jointly but with different update frequencies. We optimize the keys w.r.t the unsupervised loss and optimize the queries w.r.t both supervised and unsupervised losses. We simply do this to stabilize the training. Updating the cluster centroids with the features at the same time will lead to trivial solutions [2, 3]. Also, note that there is a strong interaction between keys (updated by unsupervised gradients in each epoch) and queries (updated by unsupervised gradients in each epoch and by supervised gradients in each batch). Every time that keys are updated, it affects the update of the queries and decoupling will ignore this interaction.
>
> If one were to train in two stages, as suggested by the reviewer: (1) the results would be sub-optimal as features and centroids are not jointly learned (i.e., see [1,2,3,4]) and (2) doing so would not allow hierarchical representation learning. For example, say there were two levels of hierarchy. If we decouple the training, we will need four stages of training: first learn node embeddings, then apply the clustering, learn the node embeddings on the new graph, cluster again. And because we limit the interaction between parameters by layer-wise pre-training, the learned features will be sub-optimal. Our integrated approach avoids this.
>
> In regards to novelty, we respectfully disagree. We introduce a memory-layer that learns representation and coarsen the graph at the same time. As far as our knowledge is concerned, ours is the first algorithm to do that. We also show that message passing is not required for learning good representation. More importantly, we show that allowing the network to decide the importance of the nodes and the neighborhood achieves better results compared to confining it to explicit neighborhoods.
>
> -  Eq. (1) and (3) are correct. The C_{i,j} soft assignment matrix is normalized row-wise to represent the probability of a query belonging to each centroid or key. These probabilities should add to one. However, there is no need for column-wise normalization since some important keys might gather large portion of information from a few queries (i.e., summation greater than one) but weak keys might collect near zero information from the queries (i.e., summation smaller than one). Similar formulation is used in baselines [5].
>
> Thanks for noticing this. It is an oversight on our end. We corrected Eq. (4) as follows:
> $$\textbf{V}^{(l)}= \textbf{C}^{(l)\top}\textbf{Q}^{(l)} \in \mathbb{R}^{n_{l+1} \times d_{l}}$$
> Where $\textbf{W} \in \mathbb{R}^{d_{l} \times d_{l+1}}$.
> We also found the same mistake with Eq. (6) and corrected it as follows:
> $$\textbf{Q}^{(0)}=\text{LeakyReLU} \left(\left[\text{LeakyReLU}(\textbf{AW}_0) \parallel \textbf{X}\right]\textbf{W}_1\right)$$
> where $\textbf{W}_0 \in  \mathbb{R}^{n \times d_{in}}$ and $\textbf{W}_1 \in  \mathbb{R}^{2d_{in} \times d_0}$.
>
> -  We revised section 3 and added the requested details. The keys are initialized randomly and are updated epoch-wise w.r.t the unsupervised loss (i.e., KL divergence). For the initialization, we also tried preloading the memory (i.e., similar to [1]) using centroids computed by K-Means over the initial node embeddings to warm-start them but did not observe any improvement over the randomly selected keys. We optimize all model parameters except the keys in each batch w.r.t Eq. (12). That means parameters are updated by the gradients of the cross-entropy loss in each batch and also are updated by the gradients of the KL divergence at the end of each epoch. Keys, on the other hand, are only updated at the end of each epoch by the gradients of the KL divergence. Updating the centroids with the same frequency as the network parameters can destabilize the training [2, 3, 4]. That is the reason why we update them epoch-wise.
>
> -  Yes, and it has been shown by others as well that it can be optimized end-to-end using mini-batch SGD [1].
>
> -  We added a hyper-parameter subsection to the revised paper. The hyper-parameters for number of keys, the number of heads, number of layers, the size of the hidden dimension, and the batch size for each and every benchmark is reported in Appendix 2, Table 3. Also, we investigate the importance of the parameters in section 4.3.
>
> -  We cited the suggested paper in the revised paper.
>
> References
> [1] Junyuan Xie et al. Unsupervised deep embedding for clustering analysis. ICML 2016.
> [2] Kaveh Hassani et al. Unsupervised multi-task feature learning on point clouds. ICCV 2019.
> [3] Mathilde Caron et al. Deep Clustering for Unsupervised Learning of Visual Features. ECCV 2018.
> [4] Elie Aljalbout et al. Clustering with Deep Learning: Taxonomy and New Methods. arXiv 2018.
> [5] Zhitao Ying et al. Hierarchical graph representation learning with differentiable pooling. NeurIPS 2018.

---

> > ### Comment · AnonReviewer1 · 2019-11-14
> > **Updated review based on rebuttal and other reviews**
> >
> > Thanks for the clarification to my questions. Based on the authors' reply and other reviews, I have changed my rating to "Weak accept". Also updated the review above.

---

> > > ### Author Response · Authors · 2019-11-15
> > > **Thank you**
> > >
> > > Thanks for the insightful feedback.

---

### Official Review · AnonReviewer2 · 2019-10-28
**Official Blind Review #2**

**Rating:** 6

**Review:**

The authors introduce a method for adding memory layers to a graph neural network which can be used for representation learning and pooling. Two variants, the MemGNN and GMN are proposed which use the memory layer. The authors evaluate their models over 7 datasets that cover classification and regression tasks. They obtain SOTA on 6/7 datasets; perform ablation analysis and introspect the clusters for chemical significance.

Overall this paper is well written and easy to read. The motivation, equations and illustrations are clear and helpful. The model is technically novel, building up from existing approaches in a progressive way. Given the generality of the approach, the impact is also likely to be high.

In order to bolster their results the authors may run their approach on a few other datasets in Wu et. al. 2018.

Minor issues:
 - Provide error bars for the tables
 - Sec 4.2 typo : “datastes”


**Experience Assessment:**

I have published one or two papers in this area.

**Review Assessment: Checking Correctness Of Derivations And Theory:**

I assessed the sensibility of the derivations and theory.

**Review Assessment: Checking Correctness Of Experiments:**

I assessed the sensibility of the experiments.

**Review Assessment: Thoroughness In Paper Reading:**

I read the paper at least twice and used my best judgement in assessing the paper.

---

> ### Author Response · Authors · 2019-11-13
> **Response to Review #2**
>
> We would like to thank the reviewer for the thoughtful comments. We address them as follows:
>
> - We added the results on Tox21 graph classification benchmark from [1] to the revised paper. We achieve a state-of-the-art AUC-ROC of 0.828 on this dataset. Per request from Reviewer #3, we also added the results on Reddit-binary graph classification benchmark (i.e., predicting community type in Reddit discussions). The GMN
> model archives state-of-the-art accuracy of 86.39% on this dataset.The results of these two datasets are reported in Appendix 3, Tables 6 and 7, respectively.
>
> - We added the error bars in Tables 2, 5, and 7 in the revised paper. These include baselines reported in [1] which contain well-documented error bars. For the other three classification baselines we did not add the error bars because they are not reported in the baseline papers.
>
> - Thanks for noticing the typo. We proof-read the paper in the revised version.
>
> References
> [1] Wu, et al.. Moleculenet: a benchmark for molecular machine learning. Chemical science 2018.

---

### Official Review · AnonReviewer4 · 2019-10-31
**Official Blind Review #4**

**Rating:** 6

**Review:**

This paper proposes a memory layer for Graph Neural Networks (GNNs) and two deep models for hierarchical graph representation learning. The proposed memory layer models the memory as a multi-head array of keys with a soft clustering mechanism and applies a convolution operator over the heads. The proposed models are experimentally evaluated on seven graph classification and regression tasks.

Generally, the paper is technically justified. The proposed technique is well motivated and properly presented. A novel clustering-convolution mechanism is proposed  for memory augmentation and graph pooling. However, there are still some rebuttal requests. 1- Some details are insufficient. For the multi-head mechanism, it is not stated clearly whether for each head an independent query is computed or a shared query is used for all heads.
2- Additionally, a related work published in NIPS 2016 should be cited and discussed.
Jack Rae et al. Scaling memory-augmented neural networks with sparse reads and writes.


**Experience Assessment:**

I have read many papers in this area.

**Review Assessment: Checking Correctness Of Derivations And Theory:**

I assessed the sensibility of the derivations and theory.

**Review Assessment: Checking Correctness Of Experiments:**

I assessed the sensibility of the experiments.

**Review Assessment: Thoroughness In Paper Reading:**

I read the paper at least twice and used my best judgement in assessing the paper.

---

> ### Author Response · Authors · 2019-11-13
> **Response to Review #4**
>
> Thank you for your supportive comments. We would like to address the points as follows:
>
> 1- We revised section 3 to add the requested details. In short: the input query is shared among the key heads. Therefore applying an input query to a layer with for example 10 key heads results in 10 assignment matrices (i.e., C). We stack the matrices to form a tensor of size $h \times n_l \times n_{l+1}$ where $h$ is the number of heads (i.e., depth in standard convolution analogy and 10 in this case), and $n_l \times n_{l+1}$ is the size of the cluster assignment matrix $C$ (i.e., height and width in standard convolution analogy). In other words, we treat each head as a separate channel. Because there is no spatial structure, we use $[1 \times 1]$ convolution to aggregate the $C_{j,j}$s across channels and therefore the convolution behaves as a weighted pooling that reduces the heads to one head. We then pass the aggregated $C$ to a non-linearity.
>
> 2- We added this work to section 2. Similar to this work, we use “content addressable memory” and use Student’s t-distribution as a kernel to measure the similarity. However, because the number of memory words is fairly small in our case (i.e., number of cluster centroids), we do not in practice need to apply the tricks mentioned in this paper for faster retrieval and updates. For example, in a few datasets that we experimented on, the maximum number of keys in each array were 32 which can be updated efficiently without requiring using approximate nearest neighbor to select the top keys to be updated.

---

### Author Response · Authors · 2019-11-13
**Paper revised to address the reviews**

First of all, we would like to thank the reviewers for their insightful and detailed comments.

We revised the paper to address these feedbacks as follows:

1- We throughly proof-read the paper.
2- We added more details to Section 3 (i.e., method).
3- We added two more datasets and showed that we achieve state-of-the-art results on them too.
4- We cited the suggested papers.
5- We revised the paper to address the missing details in all sections including Appendix.

Finally, We removed baseline results of [1] from Table 1 because we noticed they are using an augmented version of the initial features for all datasets. We and the other baselines report the results based on the following initial features:
If dataset has initial node attributes, use them as the initial node features.
If dataset does not have initial node attributes but contains node labels, use the one-hot representation of the node labels as the initial node features.
Otherwise, use a fixed length vector of ones as the initial node features.
Whereas in [1], they concatenate initial node attributes, node labels, node degree, and deterministic clustering coefficients to construct the initial node features.

References
[1] Bianchi et al. Mincut pooling in Graph Neural Networks. Arxiv 2019.

---

### Decision · Program_Chairs · 2019-12-19

**Decision:**

Accept (Poster)

**Comment:**

Four reviewers have assessed this paper and they have scored it as 6/6/6/6 after rebuttal. Nonetheless, the reviewers have raised a number of criticisms and the authors are encouraged to resolve them for the camera-ready submission.